# Incidence and risk of periodontitis in obstructive sleep apnea: A meta-analysis

Zhiqiang Zhang[1], Sitong Ge[2], Guanhong Zhai[1], Sihan Yu[1], Zhezhu Cui[1]*, Shurui Si[3], Xiang Chou[4]

1 Department of Otolaryngology, Yanbian Hospital, Yanbian University, Yanji, Jilin, China, 2 Department of Stomatology, The First Hospital Affiliated to Qiqihar Medical University, Qiqihar, Heilongjiang, China, 3 Department of Stomatology, Liaocheng People's Hospital, Liaocheng, Shandong, China, 4 Department of Infectious Disease, Yanbian Hospital, Yanbian University, Yanji, Jilin, China

* cuizz2008@163.com

## Abstract

### Introduction

At present, the possible relationship between obstructive sleep apnea and periodontitis has been reported. The link remains ambiguous and unclear. The objective of this study is to assess the association between OSA and periodontitis.

### Methods

Three databases, including Pubmed, Embase, and the Web of Science, were systematically searched to identify eligible studies that from their establishment to February 2022 for relevant studies. Subsequently, a meta-analysis was conducted to determine the relationship of pooled-effects more accurately.

### Results

A summary analysis of the 9 results from the studies covering 43,414 individuals showed a statistical association results of the between OSA and the incidence rate of periodontitis(OR = 0.52; 95% CI: 0.49–0.55; $I^2$ = 98.43%; P = 0.000). In addition, OSA patients and the risk of the population were statistically significantly associated with an increased risk of periodontitis.(OR = 1.56; 95% CI: 1.06–2.06; P = 0.00).

### Conclusions

Our results indicated that OSA may be associated with an increased risk of periodontitis. Further studies are required to confirm the link and explore the underlying mechanism of the link.

**Data Availability Statement:** All relevant data are within the paper and its Supporting Information files.

**Funding:** The author(s) received no specific funding for this work.

**Competing interests:** The authors have declared that no competing interests exist.

## Introduction

Obstructive sleep apnea (OSA) is a sleep disorder characterized by repeated episodes of complete (apnea) or partial (hypopnea) upper airway obstruction, resulting in sleep fragmentation [1] and the highest risk of death among all sleep disorders [2,3]. Current estimates of the prevalence of OSA range widely and are common in the general population worldwide. According to the latest epidemiological studies, the prevalence of OSA in western countries is 24%–49.7% in men and 9%–23.4% in women [4–6]. It has major implications for individuals, families, healthcare systems, and society. OSA is diagnosed and graded according to the frequency of Apnea-Hypopnea Index(AHI) observed during sleep, which the field of sleep medicine relies on [7]. The most common clinical manifestations of OSA are usually associated with symptoms such as snoring, inattention, excessive daytime sleepiness, fatigue, and cognitive decline, which is often associated with decreased blood oxygen saturation [8]. Untreated OSA is associated with adverse neurocognition (daytime sleepiness, reduced attention), vascular disorders (coronary heart disease, hypertension, stroke, congestive heart failure, and atherosclerosis), and metabolic disorders (hypertension, diabetes, stroke, impaired glucose tolerance, and insulin resistance) [9–12].

Periodontitis is the most common chronic infectious disease of periodontal supporting tissue and the main cause of tooth loss. The prevalence of severe periodontitis in the world is more than 10% [13,14]. About half of American adults have periodontitis, and 15% of those are severe. The current understanding of its pathogenesis is based on chronic infection caused by pathogenic microorganisms, which leads to weakened periodontal tissue function and then tooth loss [15,16]. Periodontal disease is a multifactorial disease, and the occurrence and development of periodontal disease requires the participation of multiple factors. Diabetes [17] has been confirmed as an independent risk factor in its pathogenesis. In addition, diabetes, smoking [18], cardiovascular disease [19] and genetic factors [20] have also been confirmed to be related to the incidence of periodontitis. Moreover, Periodontitis can also induce oxidative stress and systemic inflammation [21,22].

Periodontitis shares similar risk factors with OSA and is thought to have implications for systemic health due to the inflammatory response involved in its pathogenesis. Therefore, OSA may be associated with periodontitis [23,24].

However, longitudinal studies linking OSA to periodontitis have found little evidence, and population sizes are often too small to detect clinically relevant associations. In current studies, there are systematic reviews or meta-analyses on the relationship between OSA and periodontitis [25,26], but so far only one meta-analyses has been published in 2015, focusing on the correlation [25], and no studies on the incidence of periodontitis in OSA patients. In addition, most reliable and new trial results have been published since 2016. Therefore, we systematically reviewed and meta-analyzed the current population-based evidence available to determine the incidence of periodontitis in OSA and explore the association between them.

## Methods

### Protocol and registration

This study is reported according to the Preferred Reporting Items for Systematic Reviews and Meta-Analysis(PRISMA). We performed a systematic review based on a priori protocol that was registered with PROSPERO (No. CRD42021292815) [27].

### Eligibility criteria

Articles were included if they met the following criteria:

1. case-control, cross-sectional, cohort study

2. investigation into the association of OSA with risk of periodontitis

3. The incidence rate of periodontitis in OSA patients is calculated

4. (Age≥18 years) studies were included

5. Case reports, case series, or cohort studies with <20 patients were excluded. Also excluded were case reports, reviews, books, abstracts, and editorials

OSA was defined as a diagnosed paroxysmal sleep apnea syndrome that has met the conditions of providing a clear auxiliary clinical diagnosis and excluded patients with only a single diagnosis of snoring, sleep disorders, or sleep disorders.

Considering that various studies are inconsistent in the diagnosis of OSA, we mainly refer to overnight polysomnography or home sleep testing monitor—Apnea Risk Evaluation System (ARES) as the main diagnostic index. Nevertheless, we also recognize the potential importance of various clinical scales for OSA risk assessment and chose the Epworth Sleeping Score (ESS) and Berlin Questionnaire as the secondary outcome measures to provide additional insight into the association of OSA with periodontitis. we included the study with the longest follow-up or largest number of participants when more than 1 article reported data from 1 cohort or 1 health data-base. Without providing an odd-risk (OR) estimate with a corresponding 95% confidence interval (CI),were excluded from this study. The data was extracted by at least two independent investigators.

In the first stage, all potentially relevant articles were reviewed by two independent reviewers by reading titles and abstracts to decide the eligibility of studies. The full text that exclusion of non-eligible studies disagreements was obtained and assessed independently. All data were rechecked for internal consistency, if necessary, any disagreements were resolved by discussion and consultation with the third author of the review.

## Search strategy

We systematically conducted a search of the literature using the PubMed, Embase, and Web of Science databases from their establishment to February 2022 for relevant studies. A detailed and complete retrieval strategy using Obstructive Sleep Apnea, Obstructive Sleep Apnea Syndrome, Periodontitis, and periodontal disease. as the keywords is shown in S1 Table in Supplement. Additionally, in order to find other relevant articles that were not retrieved according to the search formula in the public database, we conducted a manual search of references in the included studies and of relevant reviews.

## Data extraction and quality assessment

Where possible, we extracted data on the exposures population source, study design, sample size, gender distribution, mean age and age range of study participants, and summary-level incidence of OSA and Periodontitis.

The quality of the selected papers was assessed based on the Newcastle–Ottawa scale (NOS) that has been recently recommended by systematic review practice guidelines as the most reliable instrument for conducting quality assessment of cross-sectional or cohort studies in systematic reviews. Studies with a score of points by NOS were classified as low quality (0–3 points), medium quality (4–6 points), and high quality (≥7points).

Studies with a score of 0–3 points were classified as low quality, with a score of 4–6 points were classified as medium quality, and ≥7 points were classified as high quality.

## Statistical analysis

The OR and corresponding 95%CI were extracted from each study and used to assess an association between OSA and Periodontitis. A priori, random-effects meta-analysis and meta-regression were used to quantify this heterogeneity based on previous meta-analyses [25], and we anticipated high levels of heterogeneity. We used A chi-square ($X^2$) test and I-squared ($I^2$) statistic to evaluate heterogeneity among included studies. Statistical heterogeneity was considered significant when $p < 0.10$ for $X^2$ test or $I^2 > 50\%$ [28]. Egger's regression test16 [29] and Begg's test [30] were used to statistically assess publication bias. The funnel plot was visually inspected to confirm publication bias, and we performed a sensitivity analysis by excluding one study each time and rerunning the analysis to verify the robustness of the overall results. ($P<0.05$ was considered statistically significant) conclusions were based on the use of Stata software version 16.0 (Stata Corp., College Station, Texas, US).

## Result

### Retrieved literatures

In our database, we retrieved 472 research-related literatures. Eight articles were included in this meta-analysis, including seven cohort studies and two case-control studies. This study selection process is shown in Fig 1.

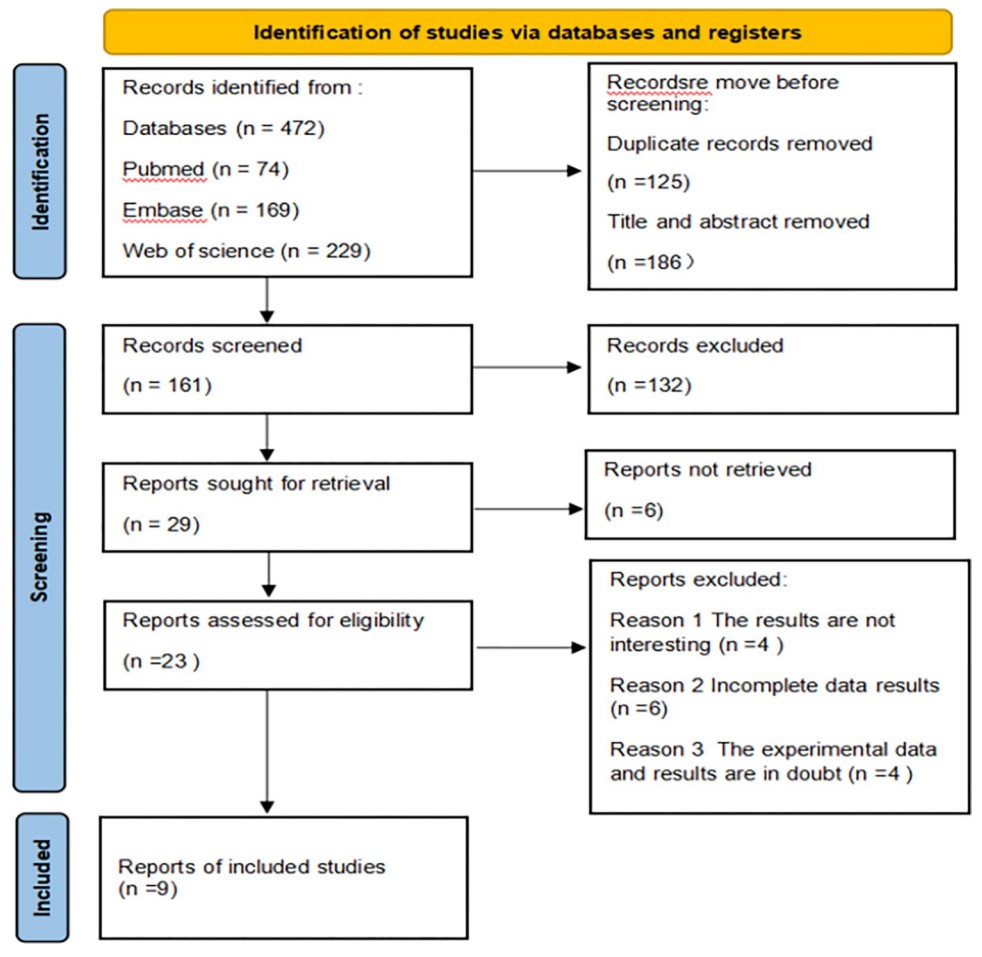

**Fig 1. Flow chart of literature retrieval.**

## Study characteristics

In total, 43,414 individuals were included in the meta-analysis. Two [31,32] of them involved a sample size of > 10,000, while seven [33–39] involved a sample size < 700.The main characteristics of the included studies are shown in Table 1.

## Quality assessment

Six articles [32–34,36–38] were considered to be of high quality and three [31,35,39] of fair quality based on the quality assessment of NOS scores. The specific assessment of NOS scores for these 9 studies is shown in Table 2.

## The incidence of periodontitis in OSA

A total of 8 articles [31–38] that analyzed the prevalence of periodontitis in patients with OSA were included in our study. The eight articles, as shown (OR = 0.33; 95% CI: 0.32–0.33; P = 0.000; $I^2$ = 98.56%) in Fig 2, describe the incidence rate of periodontitis in OSA risk patients by using dichotomous variables. Due to the high heterogeneity, we performed sensitivity analysis on the studies by sensitivity analysis was carried out by removing trials one by one to identify the heterogeneity, whether the source of the results were driven by a single study with large sample sizes. Two studies [31,32] with large sample sizes were excluded in order to reduce heterogeneity. After excluding two literatures, the inclusion study had no significant effect on OR and 95% CI(OR = 0.52; 95% CI: 0.49–0.55; $I^2$ = 98.43%; P = 0.000). Fig 3

**Table 1. Data extracted from the included studies.**

| Study | Year | Region | Sample Size | Study design | Middle Age | Periodontal assessment | OSA Diagnosis Method |
|---|---|---|---|---|---|---|---|
| Sales-Peres et al. | 2016 | Brazil | n = 108 (F = 85;M = 23) | Cross-sectional | No Apnea = 39.5 (± 1.1) With Apnea = 41 (± 1) | PD,CAL,CI | Bq,Ess |
| Gamsiz-Isik et al. | 2017 | Turkey | n = 163 (M = 122; F = 41) | Case-control | 45 | CAL,PD,BOP,PI | PSG |
| Loke et al. | 2015 | U.S.A | n = 100 (M = 91;F = 9) | Cross-sectional | 52.6 | CAL,PD,REC,PI,BOP | PSG |
| Sanders et al. | 2015 | U.S.A | n = 12,469 (M = 7473;F = 4996) | Cross-sectional | | CAL,PD,REC | ARES |
| Keller et al. | 2013 | Taiwan | n = 29,284 (M = 18,232; F = 11,052) | Case-control | 47.6(± 15.4) | PD,ABL | PSG |
| Seo et al. | 2013 | Korea | n = 687 (M = 460; F = 227) | Cross-sectional | 55.85 (± 6.63) | CAL, PD, BOP, PI, REC, GI | PSG |
| Latorre et al. | 2018 | India | n = 199 (M = 9; F = 107) | Cross-sectional | 49.9 | CAL, PD | PSG |
| Mukherjee et al. | 2021 | India | n = 250 (M = 150; F = 100) | Cross-sectional | | CAL, PD | STOP-BANG |
| Ahmad et al | 2013 | U.S.A | n = 154 (M = 61; F = 93) | Case-control | 61 | CAL, BOP, PI, REC, GI | questionnaire Self-reporte |

Probing depth(PD) clinical attachment levels(CAL) calculus index (CI) gingival bleeding index (GBI) Berlin´s Questionnaire (Bq) Epworth Sleepiness Scale (ESS) Polysomnography(PSG) female(F) male(M) alveolar bone loss(ABL) bleeding on probing(BOP) gingival recession(REC) gingival index(GI) plaque index(PI) Apnea Risk Evaluation System(ARES).

**Table 2. Quality assessment of included studies.**

| COHORT STUDIES | | | | | |
|---|---|---|---|---|---|
| First author | Year | Selection | Comparability | Outcome | Overall quality score |
| Sales-Peres | 2016 | ★★★ | ★★ | ★★ | 7 |
| Loke | 2015 | ★★★ | ★★ | ★★★ | 8 |
| Sanders | 2015 | ★★★ | ★★ | ★★★ | 8 |
| Keller | 2013 | ★★★ | ★★ | ★ | 6 |
| Seo | 2013 | ★★★ | ★ | ★★ | 6 |
| Latorre | 2018 | ★★★ | ★★ | ★★ | 7 |
| Mukherjee | 2021 | ★★★ | ★ | ★★★ | 7 |
| CASE CONTROL STUDIES | | | | | |
| Gamsiz-Isik | 2017 | ★★★ | ★★ | ★★ | 7 |
| Ahmad | 2013 | ★★★ | ★★ | ★ | 6 |

The Newcastle-Ottawa Quality Assessment Scale (NOS) (Stang, et al., 2010) was used to assess the quality of the included studies in three aspects, selection, comparison and results. The scores of cohort studies and case-control studies ranged from 0 to 9 and the higher the score, the higher the research quality. NOS scores ≥ 7, 4–6 and 0–3 represent high, medium and low quality, respectively.

shows the forest plot of periodontitis incidence in OSA risk patients, and Fig 4 shows the results of sensitivity analysis.

## OSA and risk of all-cause periodontitis

Six articles [31,32,34,35,38,39] included in the study evaluated the relationship between OSA and periodontitis. In total, a history of OSA is associated with an increased risk of periodontitis. These studies reported OSA and the risk of periodontitis. The summary results showed that OSA was associated with an increased risk of periodontitis (OR = 1.56; 95% CI: 1.06–2.06;

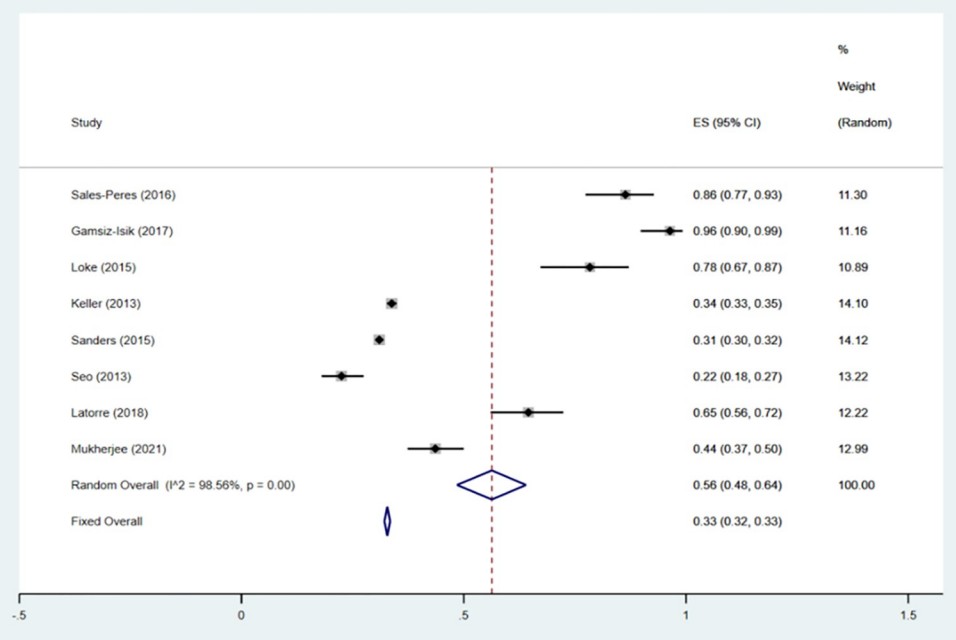

**Fig 2. Forest plot of the incidence of periodontitis in OSA according to the results of eight studies.**

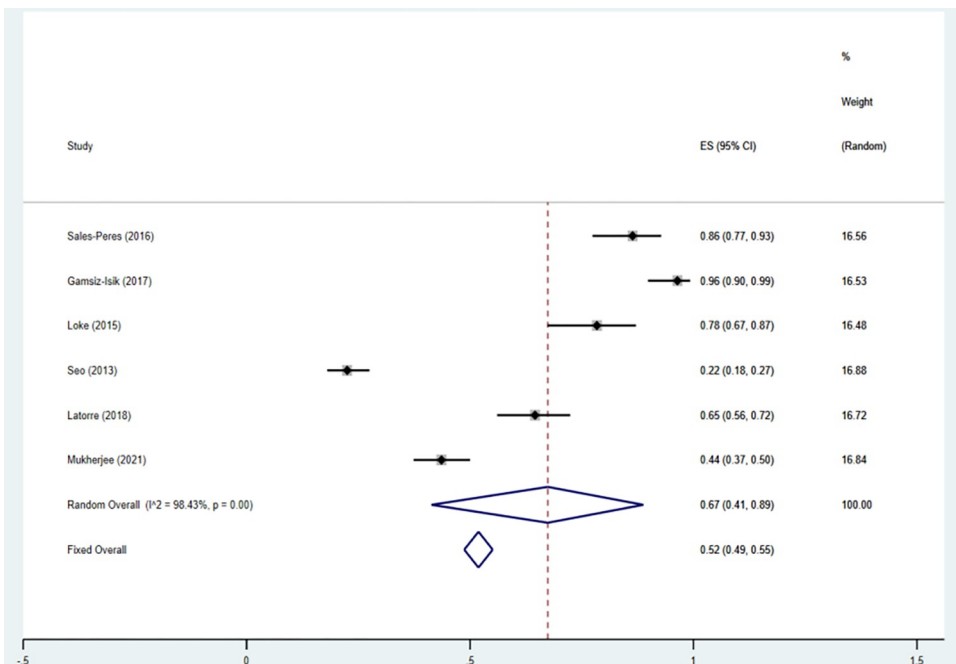

**Fig 3. Forest plot of the incidence of periodontitis in OSA according to the results of six studies.**

$I^2$ = 90.4%; P = 0.000; Fig 5), but with considerable heterogeneity across studies. It has been suggested that the origin of this heterogeneity cannot be explained by sample size (large or small) subgroup analysis or by study design (prospective or retrospective) analyses performed by omitting one individual study sequentially, as none of the studies had a significant impact on the pooled OR and 95%CI (Fig 6), and it has been suggested that the research results are

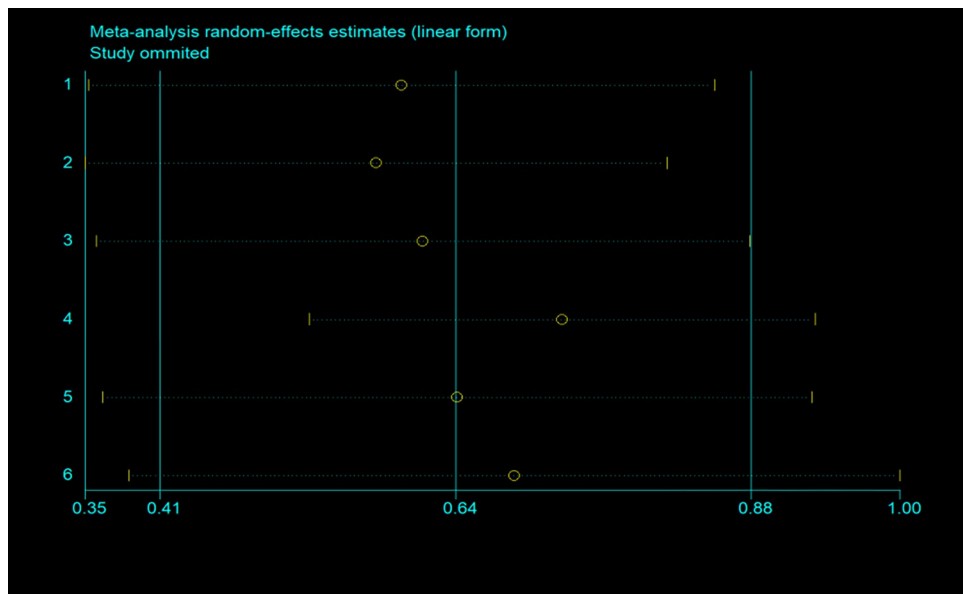

**Fig 4. Plot of sensitivity analysis by excluding one study each time and the pooling estimate for the rest of the studies (for incidence of periodontitis in OSA).**

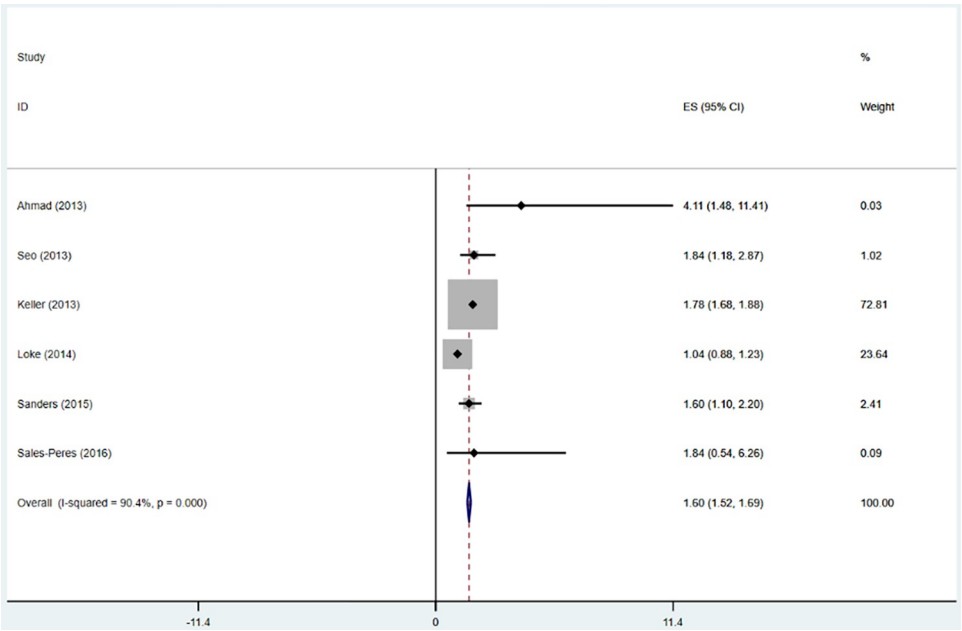

**Fig 5. Forest plot of the association between OSA and the risk of all-cause periodontitis.**

reliable. According to a visual inspection of the funnel plot analysis result, no evidence had a significant effect of publication bias (Fig 7).

## Discussion

By conducting this meta-analysis that based on 9 studies in 43,414 individuals, we performed to demonstrate the association between OSA and the future risk of Periodontitis and verified,

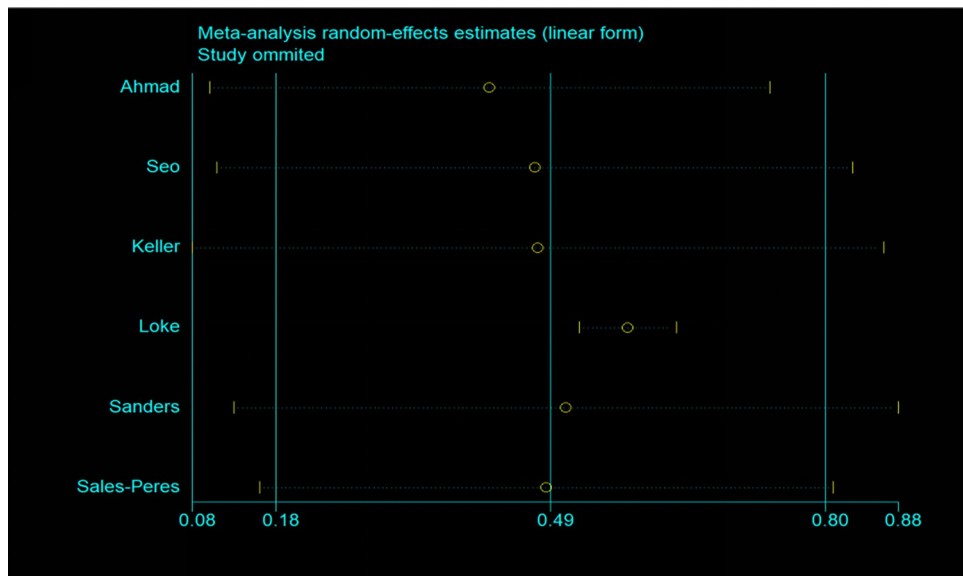

**Fig 6. Plot of sensitivity analysis by excluding one study each time and the pooling estimate for the rest of the studies (for OSA and the risk of all-cause periodontitis).**

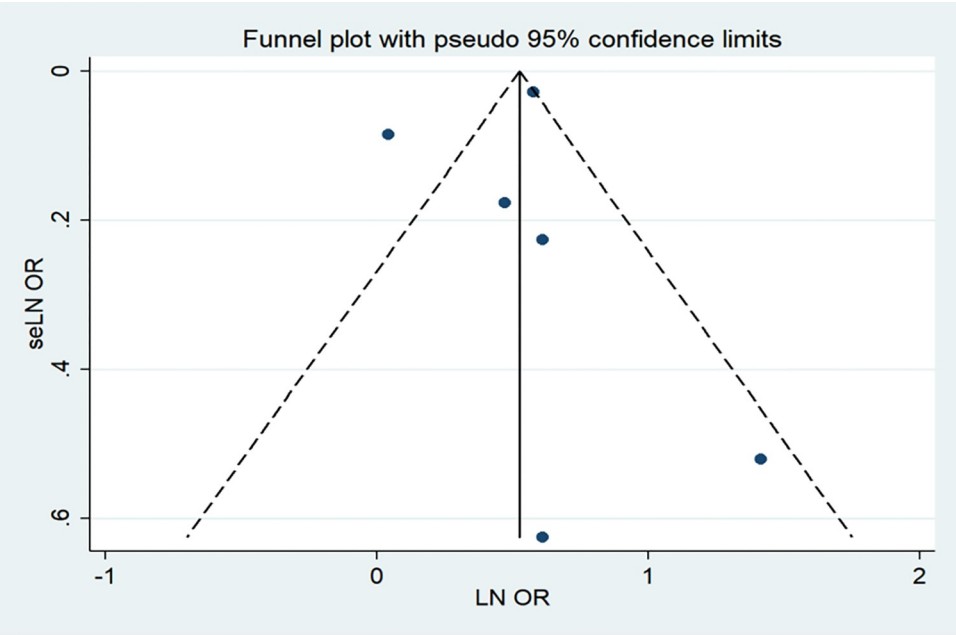

**Fig 7. Funnel plot of log relative risk vs. standard error of log relative risks (for OSA and the risk of all-cause periodontitis).**

updated, and extended a previous meta-analysis [25]. In the past, no analysis has been conducted to assess the relationship between the two by evaluating the incidence rate of Periodontitis in OSA. We believe that OSA may be a potential risk indicator for periodontitis.

Our conclusion reached above is supported by results from 2 other large population-based studies. Sanders et al. [32] screened individuals from four cities with large Hispanic/Latino populations in the U.S. between 2008 and 2011 and found that patients diagnosed with OSA were more likely to develop periodontitis and a certain dose-response relationship between them has been determined. When the AHI $\geq$ 15, the crude probability of severe periodontitis is about 7 times higher than that of AHI 0.0 (OR = 6.9, 95% Cl: 4.8, 10.0). Seo et al. [35] analyzed 3 years of follow-up data based on the Korean Genome and Epidemiology Study (KoGES), and suggested that a statistically significant correlation between OSA and periodontal risk(OR = 1.84, 95% Cl: 1.18, 2.87) after adjusting for gender, body mass index, smoking, drinking, snoring, mouth breathing, and diabetes during sleep. All these findings indicate that OSA might be a potential predictor for Periodontitis. At present, few or no meta-analyses have been conducted to confirm the relationship between OSA and periodontitis from the incidence rate. Our current meta-analysis, which included the results of 9 studies, found that the prevalence of periodontitis in patients with OSA was significantly higher than that in patients without OSA (OR = 0.52; 95% CI: 0.49–0.55; $I^2$ = 98.43%; P = 0.000). However, owing to the risk of bias, the results of this meta-systematic should be interpreted with caution. First, diagnostic criteria from different studies that were included in this meta-systematic for OSA and periodontitis could be varied. Second, some studies have conducted hierarchical research based on AHI for OSA, while others only distinguish OSA patients from non-OSA patients without hierarchical research, which has a negative impact on data extraction and result comparison. The supplied data comes from a variety of sources, some of which could originate from the clinical retrospective studies and others from public databases, which may also be an explanation for the existence of a risk of bias. Currently, relatively few studies with uniform

diagnostic standards have reported the association between OSA and periodontitis according to the incidence rate. More studies should be needed to further confirm this association.

The pathological association between periodontitis and OSA is still unclear. Although Al-Jewair et al. [25] found a statistically significant correlation between periodontal disease and OSA in a previous meta-analysis, the causal relationship between them is still controversial. However, several conclusions have been put forward to explain this relationship at present. Intermittent hypoxemia causes anoxia / reperfusion injury in OSA patients, which contributes to increased production of active oxidants and oxygen radicals and leads to the formation of other systemic inflammatory mediators in various ways, which may be the potential source of local or systemic inflammation in OSA [40].

The risk of periodontitis for patients with OSA also increases with the increase of this serum inflammatory mediator, and based on the decrease of oxygen saturation in patients with OSA, may lead to systemic inflammatory changes, including periodontal tissue [41]. It is often observed that oral breathing makes the oral mucosa dry in OSA patients, which may increase the risk of bacterial colonization and lead to higher plaque values and further increase the risk of periodontal disease [33,34].

Interestingly, periodontitis-induced systemic inflammatory responses involve systemic increases in acute-phase reactants and activated pro-inflammatory cytokines that are resident in the immune system [42], and oxidative stress is also involved in this inflammatory pathway [43]. We speculated that this similar phenomenon of oxidative stress and systemic inflammation may also be attributed to the correlation between OSA and periodontitis. In addition, the common risk factors and complications of OSA and periodontitis have also been shown to be exceedingly similar in some studies [36].

To the best of our knowledge, the present meta-analysis is the first that attempts to assess and summarize the relationship between OSA and periodontitis risk from the perspective of incidence rate and updates the previous studies of Al-Jewair et al [25] from the perspective of risk factors. Sensitivity analysis has indicated the stability of the main results. Nevertheless, the results of the present meta-analysis are subject to several limitations that should be considered with a rigorous attitude. In various studies, different criteria were used for the diagnosis of OSA and periodontitis. This may affect the selection of sample group and make the included studies show heterogeneity that exists between individual studies and may restrict our final conclusions. Additionally, at present, the number of only 9 studies eligible for inclusion was small in this meta-analysis. This study did not systematically discuss the relationship between OSA and periodontitis from the perspective of dose-response or cause-effect. The results and conclusions of this meta-analysis were only preliminary and require further analysis, which will necessitate more research results on the relationship between OSA and periodontitis. Even so, despite these limitations, our analysis still has extraordinarily significant significance. This study reveals a potential association between periodontitis and OSA, contributing to identifying the susceptible group of periodontitis or periodontal disease. Further exploration of the mechanism of the association between OSA and periodontitis is required, which may enable to provide new strategies for the treatment and prevention of periodontitis.

## Conclusions

Our present study finds that a relationship exists between OSA and periodontitis. However, published data on the relationship between OSA and periodontitis is still insufficient. Further studies are needed to understand the underlying mechanisms of association, especially studies demonstrating the link between dose-response and cause-effect.

## Supporting information

**S1 Checklist. PRISMA checklist.**
(DOCX)

**S1 Table. Literature retrieval strategy.**
(DOCX)

## Acknowledgments

We would like to thank all the authors who made outstanding contributions to this study.

## Author Contributions

**Conceptualization:** Zhiqiang Zhang, Sitong Ge, Zhezhu Cui, Shurui Si, Xiang Chou.

**Data curation:** Zhiqiang Zhang, Sitong Ge, Guanhong Zhai.

**Formal analysis:** Zhiqiang Zhang, Sitong Ge, Guanhong Zhai, Sihan Yu.

**Methodology:** Zhiqiang Zhang, Sitong Ge, Guanhong Zhai, Sihan Yu, Shurui Si.

**Supervision:** Zhezhu Cui, Xiang Chou.

**Visualization:** Sitong Ge, Guanhong Zhai, Sihan Yu.

**Writing – original draft:** Zhiqiang Zhang.

**Writing – review & editing:** Zhiqiang Zhang, Sitong Ge, Sihan Yu, Zhezhu Cui.

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
