## [Decision Letter · Decision Letter 0]

15 Jun 2022

PONE-D-22-09585Incidence and risk of periodontitis in obstructive sleep apnea:a meta-analysisPLOS ONE

Dear Dr. Cui,

Thank you for submitting your manuscript to PLOS ONE. After careful consideration, we feel that it has merit but does not fully meet PLOS ONE’s publication criteria as it currently stands. Therefore, we invite you to submit a revised version of the manuscript that addresses the points raised during the review process.

We look forward to receiving your revised manuscript.

Kind regards,

Federica Provini

Academic Editor

PLOS ONE

Journal Requirements:

Reviewers' comments:

Reviewer's Responses to Questions

**Comments to the Author**

1. Is the manuscript technically sound, and do the data support the conclusions?

Reviewer #1: Yes

2. Has the statistical analysis been performed appropriately and rigorously? 

Reviewer #1: Yes

3. Have the authors made all data underlying the findings in their manuscript fully available?

Reviewer #1: Yes

4. Is the manuscript presented in an intelligible fashion and written in standard English?

Reviewer #1: Yes

5. Review Comments to the Author

Reviewer #1: The study presents a current and relevant theme. I suggest the following revisions.

In the Abstract, Introduction, I think the aim of the study was to carry out a systematic review with meta-analysis on the association between OSA and periodontitis. I suggest that the authors rewrite this Abstract topic.

In the Introduction of the manuscript, the theme is well founded. However, I propose that the association between gingival dryness in patients who snore and/or present OSA be reported, which may trigger periodontal problems. There are studies that report this association. I think this description helps to substantiate the possible association between periodontitis and OSA.

Methodology: The methodology is very well described. I propose that authors present the elaborate search key.

Discussion: This topic is very well described. I suggest that the authors discuss the fact that most of the selected studies in this manuscript, belonged to the Asian continent. Why has the Asian continent studied this topic more? I think it's important to discuss.

6. PLOS authors have the option to publish the peer review history of their article (what does this mean?). If published, this will include your full peer review and any attached files.

**Do you want your identity to be public for this peer review?** For information about this choice, including consent withdrawal, please see our Privacy Policy.

Reviewer #1: **Yes: **Junia Maria Serra-Negra

---

## [Author Response · Author response to Decision Letter 0]

21 Jun 2022

Response to editors’ and reviewers’ comments on the manuscript submitted by Zhang et al, “Incidence and risk of periodontitis in obstructive sleep apnea: a meta-analysis”. (Manuscript ID: PONE-D-22-09585)

We appreciate the editors and reviewers for their constructive and valuable comments. We have revised our manuscript according to the editors’ and reviewers’ comments, questions, and suggestions. In the event that we missed any one of the comments please let us know. This document includes our responses to editors’ and reviewers’ comments point by point, and the revised portion are marked in RED color in our revision. 

Comments from Editor:

Comment 1: Journal Requirements:

Answer：

Thank you for your kind and thoughtful advice. We have made detailed revisions and adjustments to the formatting of our manuscript in accordance with PLOS ONE's style requirements. Although we are very attentive, there may be some omissions or deficiencies, please let us know after reviewing, and we will spare no effort to improve.

Comments from Reviewer 

Comment 1: The study presents a current and relevant theme. I suggest the following revisions.

Answer：Thank you for the effort you put into our manuscript. We appreciate your constructive and valuable comments. We have revised our manuscript according to your comments point by point below.

Comment 2: In the Abstract, Introduction, I think the aim of the study was to carry out a systematic review with meta-analysis on the association between OSA and periodontitis. I suggest that the authors rewrite this Abstract topic.

Answer：Thank you for your inspiring comment. The topic of the Abstract has been rewritten as per your comment. See Page 2, Line 2~4.

Change in text: The possible relationship between obstructive sleep apnea (OSA) and periodontitis is contradictory. Thus, the purpose of this study was to evaluate the association between OSA and periodontitis.

Comment 3: In the Introduction of the manuscript, the theme is well founded. However, I propose that the association between gingival dryness in patients who snore and/or present OSA be reported, which may trigger periodontal problems. There are studies that report this association. I think this description helps to substantiate the possible association between periodontitis and OSA.

Answer：Thank you for your great insight. Snoring and OSA can lead to gingival dryness, which in turn leads to periodontitis, which is also a common clinical phenomenon, so we deeply accept this view. We have further searched the literature to supplement and improve. See Page 4, Line 46~48.

Change in text: In addition, gingival dryness, smoking, cardiovascular disease and genetic factors have also been confirmed to be related to the incidence of periodontitis [18-20].

Comment 4: Methodology: The methodology is very well described. I propose that authors present the elaborate search key.

Answer：Thank you for your praise. Our detailed retrieval strategy is attached, which is to reduce the length and repeatability of the manuscript. Interested readers can download and view it. See S1 Table in Supplement.

Comment 5: Discussion: This topic is very well described. I suggest that the authors discuss the fact that most of the selected studies in this manuscript, belonged to the Asian continent. Why has the Asian continent studied this topic more? I think it's important to discuss.

Answer：Thank you for this interesting discovery. To be honest, we didn't realize that Asia had the most studies, if you hadn't reminded us. Under your reminder, we have carefully reviewed the included studies and tried to find a more comprehensive explanation. We tried to search the literature to prove this point, but failed to find a suitable one. Therefore, we raised this question in the limitations of this manuscript. See Page 15, Line 280~282.

Change in text: In addition, the included studies focused more on Asian populations, and more research is needed to explore this association in populations of other continents.

---

## [Decision Letter · Decision Letter 1]

7 Jul 2022

Incidence and risk of periodontitis in obstructive sleep apnea:a meta-analysis

PONE-D-22-09585R1

Dear Dr. Cui,

We’re pleased to inform you that your manuscript has been judged scientifically suitable for publication and will be formally accepted for publication once it meets all outstanding technical requirements.

Kind regards,

Federica Provini

Academic Editor

PLOS ONE

---

## [Editor Report · Acceptance letter]

12 Jul 2022

PONE-D-22-09585R1 

Incidence and risk of periodontitis in obstructive sleep apnea:a meta-analysis 

Dear Dr. Cui:

I'm pleased to inform you that your manuscript has been deemed suitable for publication in PLOS ONE. Congratulations! Your manuscript is now with our production department. 

Kind regards, 

on behalf of

Dr. Federica Provini 

Academic Editor

PLOS ONE